# Laboratory experiments in ocean alkalinity enhancement research

Maria D. Iglesias-Rodríguez[1,2], Rosalind E.M. Rickaby[3], Arvind Singh[4], James A. Gately[1,2]

[1] Department of Ecology, Evolution, and Marine Biology, University of California, Santa Barbara; Santa Barbara, CA 93106, U.S.A.
[2] Marine Science Institute, University of California, Santa Barbara, CA 93106, U.S.A.
[3] Department of Earth Sciences, University of Oxford, Oxford, U.K.
[4] Physical Research Laboratory, Navrangpura, Ahmedabad 380 009, India.

*Correspondence to*: M. Débora Iglesias-Rodríguez, iglesias@ucsb.edu

**Abstract.** Recent concern about the consequences of continuing increases in atmospheric $CO_2$ as a key heat-trapping agent (USGCRP, 2017; IPCC, 2021) have prompted ocean experts to come together to discuss how to provide science-based solutions. Ocean alkalinity enhancement (OAE) is being considered not only as an ocean carbon dioxide removal (CDR) approach, but also as a potential way to mitigate ocean acidification. Over the last two decades, inter-laboratory comparisons have proven valuable in evaluating the reliability of methodologies associated with sampling and analysis of carbonate chemistry parameters, which have been routinely used in ocean acidification research. Given the complexity of processes and mechanisms related to ecosystem responses to OAE, consolidating protocols to ensure compatibility across studies is fundamental for synthesis and upscaling analysis. This chapter provides an overview of best practice in OAE laboratory experimentation and facilitates awareness of the importance of applying standardized methods to promote data re-use, inter-lab comparisons, meta-analysis and transparency. This chapter provides the reader with the tools to (1) identify the criteria to achieve the best laboratory practice and experimental design; (2) provide guidance on the selection of response variables for various purposes (physiological, biogeochemical, ecological, evolutionary) for inter-lab comparisons; (3) offer recommendation for a minimum set of variables that should be sampled and propose additional variables critical for different types of synthesis and upscaling; and (4) identify protocols for standardized measurements of response variables. Key recommendations include ensuring reproducibility through appropriate experimental design and replication, assessing alkalinity thresholds for secondary precipitates for each experimental approach and condition, using recommended targets of alkalinity (3000-4000 μmol $kg^{-1}$) and levels exceeding these concentrations to mimic responses at the site of deployment/non equilibrium and use intermediate alkalinity levels to identify potential nonlinear responses, and establish the appropriate experimental design to address questions at specific levels of organization (chemical, physiological, molecular) and assuming different scenarios (e.g., mimicking impacts at the site of deployment in a non-equilibrated system versus steady state scenarios in an equilibrated system.

## 1. Introduction

Laboratory studies need to be reproducible, consistent and transparent (Box 1) to provide the scientific community and regulators with useful information to move the field forward and facilitate the development of safe guidelines. Based on numerous modeling studies, ocean alkalinity enhancement (OAE) appears to be a promising ocean carbon dioxide removal (CDR) approach, with the likely beneficial side effect of mitigating ocean acidification (Burt et al., 2021; Hartmann et al., 2023; NASEM, 2022; Wang et al., 2023). Laboratory experiments are urgently needed to determine the CDR potential of various OAE methods as well as OAE impacts at various levels of biological organization (ecological, physiological, biochemical, molecular). The emerging empirical studies are offering insight while revealing gaps in our knowledge of the mechanisms governing OAE and its effect on marine biota (e.g., Ferderer et al., 2022; Gately et al., 2023; Yang et al., 2023). For example, the conditions preventing or limiting the formation of secondary precipitates and the pros and cons of various alkali are still under debate. Given that empirical work on OAE is still in its infancy and that some of the assumptions based on modeling studies remail untested, this chapter is an evolving document that will be updated as the OAE community continues to release results.

Laboratory manipulations allow making observations in a highly controlled environment using model species or subsets of populations (selected species or populations). Results are generally considered highly reproducible (Box 1) and therefore laboratory manipulations are viewed as a necessary step to either generate hypotheses to test in the field or vice versa, when field experimentation is an option. Under the latter, field observations guide the laboratory experiments to validate field results in well-known systems and under tightly controlled conditions.

A number of approaches – batch, semi-continuous and continuous cultures – have been used to address diverse OAE settings (e.g., at the point of deployment, under steady state conditions, air- versus non air-equilibrated seawater) and various biological scenarios (specific stages of growth, life cycle, and to explore abrupt/short term versus long term responses to manipulations). In some cases, specific stages during the life cycle of organisms can be selected (for example, larval versus adult stage; sexual versus asexual phase). Time series laboratory experiments are less restricted than mesocosm experiments with regards to the duration of experiments because they tend to be 'cleaner', with relatively low bacteria numbers and generally without biological confounding factors (viruses, predation, competition for resources, etc.). Therefore, the cause–effect relationships are easier to elucidate as conditions and organisms can be tested in relative isolation, and there is the possibility of extensive replication.

The main limitation of laboratory experiments is that the dynamic phenomena occurring in the natural environment cannot be captured in the laboratory and, therefore, results may not be applicable to real life scenarios. For example, in laboratory experiments the influence of mixing processes, conditions governing particle flocculation or the linkage to higher levels of biological organization (e.g., predation) are difficult to discern (see Forbes and Calow, 2002; Martin et al., 2014). Portable lab experiments, such as deck incubations abord research vessels or outdoor incubations, with some influence from the local environment (e.g., diurnal alterations of light, water flow through from the coast to maintain in-situ temperature) as well as community-level mesocosm experiments are the conduit to field manipulations. These large-scale community experimental tanks address the importance of the physico-chemical conditions, space, density-dependent effects, biotic interactions and complexity of natural environments in their response to OAE manipulations\buffering, or boosting, the direct effects of environmental stress on organisms (Paiva et al., 2021).

This chapter provides best practice guidelines in OAE laboratory experimentation and offers recommendations to enable data re-use, inter-lab comparisons, and transparency. We offer recommendations regarding (1) the criteria to achieve the best laboratory practice and experimental design; (2) the selection of response variables for various purposes (physiological, biogeochemical, ecological, evolutionary) for inter-lab comparisons; (3) a minimum set of variables that should be sampled and additional variables critical for different types of synthesis and upscaling; and (4) protocols for standardized measurements of response variables.

### 2. Lessons learned from ocean acidification research

The rich insights obtained in ocean acidification research are key to supporting OAE studies. However, as crucial as it is to follow guidelines when designing laboratory experiments, it is equally important to acknowledge that there may be potential confounders and challenges that may not be accounted for in the guidelines. Being able to conduct quantitative laboratory intercomparisons, including interspecies comparisons, will be critically dependent on identifying recommendations regarding experimental design, sample collection and data analysis. Important considerations include the source of alkalinity, rate of alkalinity addition, testing air-$CO_2$-equilibrated *versus* non-equilibrated seawater, and the effect of ancillary variables (e.g., temperature) in multifactorial experiments which are known to yield complex and variable results (e.g., see the interactive effects of ocean acidification and warming - Harvey et al., 2013). The guidelines provided in this chapter should significantly improve the quality and impact of the OAE research, which is required to meet the identified societal need for research on OAE and other types of ocean CDR (NASEM, 2021).

An exploration of procedures, patterns and challenges associated with ocean acidification research has offered ideas on how to design rigorous and reproducible laboratory experiments that enable measuring and monitoring carbonate chemistry shifts and biological responses to ocean acidification (Cornwall and Hurd, 2016). Cornwall and Hurd (2016) reported that 95% of the experimental work between 1993 and 2014 had interdependent or lacked replication in clearly defined treatments, or did not report sufficient methodological detail. More broadly,

results from Wernberg et al. (2012) from marine climate change experiments between 2000 and 2009, reported that ~49% of the experiments had identifiable issues with their experimental procedures, and 91% of the experiments reported showed a lack of treatment replication or pseudo-replication. Amongst the studies, 9% included extreme/unrealistic treatments of temperature or pH far beyond worst case scenario projections (Wernberg et al., 2012) although 'extreme' pH/alkalinity conditions may prove useful to define thresholds of tolerance and upper limits of alkalinity enhancement, and to understand underlying physiological mechanisms of acclimation to alkalinization. While the urgent need for field trials requires careful consideration of treatment levels, in order to maximize the insight gained from OAE experiments, testing conditions outside the year 2100 IPCC $CO_2$ emission scenarios are encouraged. These conditions outside worst case scenario projections will further our knowledge on the mechanisms governing biological (e.g., shell production) and abiotic (e.g., particle aggregation, secondary precipitation) responses to applied chemical CDR.

Like in ocean acidification research, careful attention should be given to the advantages and disadvantages that concern the choices of dissolved inorganic carbon species to measure, and how error propagation will affect the calculated parameters (Martz et al., 2015). Moreover, dissolved organic matter (DOM) is known to contribute to alkalinity (Kim and Lee, 2009; Koeve et al., 2010) although the presence of strong acidic groups in organic matter can decrease net alkalinity (Hu, 2020; Middelburg et al., 2020). Depending on the type of system under investigation, attention should be paid to whether to apply titration alkalinity (typically used in ocean studies) versus the charge balance approach (often used in freshwater systems, with high concentrations of dissolved organic matter) (see Middlebufg et al., 2020). Results from ocean acidification mesocosm experiments focused on phytoplankton revealed that nutrient-limited communities appeared to be more responsive to changing carbonate chemistry than those having access to high inorganic nutrient concentrations (see Paul et al., 2015; Sala et al., 2015; Bach et al., 2016). These observations indicate that trophic state might play a role in the susceptibility of organisms to the changes in carbonate chemistry driven by alkalinization. Also, competition between species has been found to be altered under various carbonate chemistry conditions (see Kroeker et al., 2013a), which merits a focus on experiments that address preferential selection of taxonomic groups under different alkalinity conditions. Although applying nutrient-limiting conditions is experimentally challenging, understanding how species succession and community composition might respond to alkalinization could in part be addressed in a laboratory context.

While it is fairly straightforward to determine how individual changes in parameters influence chemical and biological responses, understanding impacts of multiple parameters [e.g., increased alkalinity and warming, increased alkalinity and resource availability (nutrients, light, prey)] can be challenging as they can interact in complex ways. Indeed, ocean acidification research revealed antagonistic, synergistic, and additive responses when studying ocean acidification and warming (Byrne and Przeslawski, 2013; Kroeker et al., 2013b; Harvey et al., 2015; Pistevos et al., 2016). Identifying tipping points and interactive effects when other parameters (e.g., temperature) are altered in seawater, in addition to alkalinity, is critical given the capacity of these parameters to drive (otherwise unpredictable) shifts in species abundances, biodiversity and community composition, physiological outputs, survival, and reproduction (Crain et al., 2008; Darling and Côté, 2008; Galic et al., 2018).

**Box 1.** **Criteria for best laboratory practice**

· *Reproducibility*. From the emerging OAE research (e.g., regarding the formation of secondary precipitates - see Montserrat et al., 2017 versus Fuhr et al., 2022; and Moras et al., 2022) and the ocean acidification literature (e.g., see Ridgwell et al., 2009), we have learned that similar approaches can lead to conflicting and unresolved outcomes. Without appropriate reporting of sample collection, methodology and data processing, it is challenging to re-analyze the data and reconcile the discrepancies. As the field emerges and evolves, it will be required to reevaluate early experiments and possibly re-analyze results with updated protocols.

· *Defining inclusion and exclusion criteria.* In order to reduce confounding covariates, attention must be paid to factors affecting flocculation, aggregation of particles (e.g., possibly impacted by dissolved organic matter increases after phytoplankton blooms), fluctuations in temperature, which affect mineral dissolution and precipitation rates, and biological and physiological properties, including stage during the life cycle, trophic state, and seasonality, that affect the susceptibility of organisms to OAE (e.g., see Vandamme et al., 2015; Subhas et al., 2022).

· *Establishing experimental controls.* In OAE experimental designs, controls must be appropriately selected. These could include seawater without added alkalinity, seawater ± nutrients/food, treatments

with and without the organisms tested. When mineral dissolution is too slow, an alternative analog that reproduces the basic chemistry is encouraged (for example, the use of salts and alkali; e.g., $CaCl_2$ and $Na_2CO_3$ to mimic the effect of limestone-based mineral dissolution). Controls could also contain an alternative form of alkalinity that alters the seawater carbonate chemistry solely, without adding carbon or trace metals (e.g., NaOH).

· ***Basic biological responses.*** Studies on organisms' physiological responses (e.g., growth, respiration, size, reproduction, photosynthesis and calcification) are recommended. These responses can be measured directly; for example, as uptake rates of solutes using traditional assays, mass spectrometric methods for indirect assessment of changes in elements, or molecular responses using markers of functional processes. Rates of growth and calcification can also be measured by changes in dry mass or buoyant mass in many types of organisms, especially in macroinvertebrates and macroalgae (see Dodge et al., 1984; Davis, 1989; Sanders et al., 2018). For organisms that undergo development one must determine which stage of development (e.g., larval vs adult; vegetative vs gamete stage) to target. Also, when altering more than one parameter, particular attention must be paid to potential confounding effects. Multi-factorial experiments can be used to explore the weight of each parameter.

### 3. Seawater media preparation and manipulation of carbonate chemistry

The different steps in experimental design are outlined in Table 1. The process starts with natural or artificially made seawater with or without nutrient additions. One must consider whether adding nutrients/food/prey is required; for example, whether exploring OAE impacts is intended in conjunction with specific scenarios, e.g., nutrient fertilization, specific stages of growth or population development, and the extent to which nutrient additions or any other basic manipulation of the environmental conditions might impact the interpretation of results. For OAE manipulations where sterilization is required for the experimental set up, autoclaving is discouraged given the alterations in carbonate chemistry, including loss of $CO_2$, leading to a decrease in dissolved inorganic carbon and alterations in alkalinity (increase with increasing salinity/decrease with precipitation of carbonate) triggered by autoclaving. Instead, filter-sterilization of seawater through small pore size filters (e.g., 0.22 µm filters) is required to remove particles and most bacteria, and produce the stock media where different manipulations are applied to create different alkalinity treatments.

There are several approaches to simulating the addition of alkalinity that capture different components of any manipulation experiment. The first approach could be testing the impact of instantaneous addition of alkalinity to seawater to mimic the impact on seawater chemistry and ecosystems at the point of deployment. The second involves aeration and equilibration with the atmosphere to explore the physico-chemical response to a staeady state/ equilibrated scenario. In the latter instance, the medium is aliquoted out to the experimental vessels/tanks where aeration is applied to promote air equilibration. Monitoring carbonate chemistry through time enables determining when equilibration of seawater with air occurs.

**Table 1. Experimental considerations for OAE experimentation.**
**Medium preparation**: the seawater can be obtained from coastal or open ocean sites. Filtered seawater or, when appropriate (e.g., when growing autotrophic organisms), seawater supplemented with nutrients; for example, using f/2 or variations of f/2 media (see Guillard and Ryder, 1962) will be used for growing organisms. Seawater media can also be prepared from artificial recipes (e.g., Aquil medium; Morel et al., 1979) when specific compounds or elements need to be altered in seawater. Media must be sterilized by filtration rather than through autoclaving and nutrients can be added, typically from stock solutions. When possible, moderate aeration should be applied. Types of alkali include adding pulverized mineral directly to the media and promote dissolution physically (e.g., by stirring); dissolving the mineral separately and filter out any particles remaining in the media before experimentation; dissolving salts to mimic the chemistry of the dissolved alkali (e.g., to mimic limestone dissolution, dissolve $CaCl_2$ and $NaCO_3$, which result in higher dissolution rates); and adding liquid alkali such as NaOH. Establishing time series prior to the experiment to determine time frames regarding length of experiment, frequency of sampling, etc. is recommended. **Experimental design**: in addition to optimizing reproducibility by designing enough replication and test the reproducibility of the method, researchers should remain engaged with respect to protocols and experimental design to avoid artifacts and undesirable side effects of methodology. When possible, ensure equilibration of seawater gasses with air and define experimental time frames to test impacts under conditions representative of the site of deployment (where limited gas exchange occurs) and those representative of steady state/equilibrated conditions. Although most laboratory experiments address short term impacts, chronic

effects can be tested in long term incubations. **Sampling and analysis**: the parameters to be considered should allow inter-lab comparisons, address functional properties of organisms (e.g., calcification, silicification, particulate organic carbon) and fulfill needs to improve model parameterizations. It is important to establish well defined time windows for sampling as well as frequency of sampling to capture physical, chemical and biological properties of the studied system. It is advisable to limit the time of sample storage to minimize observations that might confound interpretation of results (e.g., reverse weathering during storage). Stock solutions (e.g., nutrient and alkalinity solutions) must be stored in the appropriate vessels to avoid contamination from leachates coming out of the vessel itself (e.g., silicate contamination from solutions stored in borosilicate containers). Detection limits and accuracy and precision should be offered for each protocol.

| Medium preparation | Experiment design | Sampling and analysis |
|---|---|---|
| **Natural/artificial seawater**<br><br>**Filter sterilization (e.g., 0.22 um)**<br><br>**+/- nutrient addition**<br><br>**+/- aeration**<br><br>**Type of alkalinity treatments**<br>• Pulverized mineral<br>• Pre-dissolved mineral<br>• Dissolved salts<br>• Liquid alkali<br><br>**Pre-equilibrated vs non-equilibrated seawater with air phase**<br>• Carbonate chemistry<br>• Flocculation/aggregation<br>• Biology | **Best actions to maximize confidence**<br>• Within study replication and pseudo-replication<br>• Coordinated networks (teams sharing progress to decide on best protocols)<br><br>**Preliminary time series of TA and carbonate chemistry**<br>• Define experimental time frames<br>• Assess TA upper limits<br>• Expand the upper limits to address impacts at site of deployment<br><br>**Abrupt vs chronic biology impacts**<br>• Short-term tests (acclimation)<br>• Long-term experiments (adaptation) | **Criteria for key parameters**<br>• Inter-lab comparisons<br>• Functional properties<br>• Model parameterization<br><br>**Sampling frequency and timing**<br>• Select time window for sampling<br>• Identify sampling frequency that captures key chemical, physical or biological features<br><br>**Limit storage to minimize artifacts**<br><br>**Identify and report key analytical parameters affecting error**<br>• Detection limits<br>• Measurement accuracy/precision<br>• Identify any impact of experimental design on uncertainties |

### 3.a Sources of alkalinity

As yet, it is unclear what the optimal method or source of alkalinity enhancement may be in order to simulate the desired chemistry in seawater media. Proposed sources of alkalinity include silicate minerals (olivine, basalt), brucite, limestone and its derivatives (quicklime and portlandite), NaOH and mine tailings (NASEM, 2021; Nawaz et al., 2023). Given the slow dissolution kinetics of the minerals, generating alkaline solutions artificially is acceptable. For example, Gately et al. (2023) simulated alkalinity enhancement via a limestone-inspired solution by adding $Na_2CO_3$ and $CaCl_2$ or its hydrated form ($CaCl_2H_4O_2$) to seawater. Adding $Na_2CO_3$ raises TA and DIC in a 2:1 ratio, with 2 moles of TA added by 2 conservative $Na^+$ ions in $Na_2CO_3$, and 1 mole DIC added by $CO_3^{2-}$. $CaCl_2$ does not raise alkalinity because it adds equal amounts of positive and negative conservative charge to the solution from $Ca^{2+}$ and 2 x $Cl^-$. However, it does raise the calcium in solution and therefore the saturation state of the seawater with respect to $CaCO_3$.

Many possibilities for solid or liquid alkalinity additions are being considered (see chapter 3). While adding minerals as precursors of alkalinity can provide a source of potentially beneficial nutrients (e.g., silicate, iron, magnesium) (Hartmann et al. 2013), the possible toxic effect of metals leached out of minerals, an example being nickel (Ni) leached from olivine (Montserrat et al., 2017) is of concern. The use of NaOH is currently gaining attention given that its environmental footprint is perceived as smaller than the mining of alkaline minerals, which necessitate an expansion of mining operations, transportation, and industrial processing, which are energetically costly and can lead to air pollution. Additionally, the amount of Na added to seawater is very small relative to the large background of NaCl in seawater.

The addition of NaOH and other forms of alkalinity to seawater cause initial spikes in pH and a drop in aqueous $CO_2$ that can be balanced to a steady state via bubbling with air (Table 1). Determining abiotic and biotic

responses to the initial spikes in pH and drops in $CO_2$ is an important step in addition to understanding responses
under steady-state conditions. It may be that large manipulations of alkalinity are needed to elicit a measurable and
reproducible response, and the required alkalinity concentrations will be refined with more detailed modeling but,
based on current information, proposed targets for alkalinity manipulations are 3000-4000 $\mu mol\ kg^{-1}$ (Renforth and
Henderson, 2017). ~4000 μmol/kg is the concentration of alkalinity expected at locations in the ocean where
alkalinity is initially added, and ~3000 μmol/kg is the concentration of alkalinity expected once ocean circulation
has dispersed the alkalinity over a larger area (Renforth and Henderson, 2017). Alkalinity thresholds for the
formation of precipitates will need to be determined for each experimental approach and condition. It is however
recommended that researchers consider using alkalinities exceeding the recommended targets, and utilize
intermediate treatments (e.g., 2000, 4000, 7000 μatm/kg seawater) rather than just low/high treatments, in order to
identify potential nonlinear and even parabolic responses. This approach led to important and unexpected outcomes
in ocean acidification research (e.g., Ries et al., 2009).

**3.b Impacts of impurities/metal leachates**
An important consideration in OAE studies is the impact of metals leached from dissolving minerals and
their ecotoxicological potential on marine organisms. For example, although some elements (e.g., Fe and Mg)
leached out of minerals could be beneficial micronutrients, the potentially toxic effect of metals such as nickel (Ni)
(Montserrat et al., 2017), leached from olivine, is of concern. Diverse responses have however been reported with
respect to Ni and it appears that some cyanobacteria rely on Ni more than other photosynthetic organisms (see
Dupont et al., 2008, 2010; Ho, 2013). A recent laboratory study testing olivine leachates (containing Si, Ni, Mg, Fe,
Cr and Co) in phytoplankton revealed either positive or neutral physiological short term responses in all treatments
(Hutchins et al., 2023). However, one should consider the role of long-term experiments to examine organismal and
population adaptation of metal exposure as well as potential bioaccumulation and biomagnification impacts in
consumers.

Another important consideration is the effect of pH on metal speciation as pH and a change in the
concentration of $OH^-$ and $CO_3^{2-}$ ions can affect the solubility, adsorption, toxicity, and rates of redox processes of
metals in seawater thus altering the interactions of metals with marine organisms (Millero et al., 2009). When
dissolving minerals in seawater one must consider nonstoichiometry and incomplete dissolution perhaps as a result
of dissolution of impurities, precipitation of secondary minerals, or preferential leaching of elements from the
mineral surface (Brantley, 2008, NASEM, 2021). The formation of secondary precipitates has been observed in
several studies exploring the dissolution of olivine (Fuhr et al., 2022), and limestone derivatives (Moras et al., 2022;
Gately et al., 2023; Hartmann et al., 2023). Using an alkaline solution rather than reactive alkaline particles has been
recommended to reduce carbonate precipitation unless seawater critical supersaturation levels are exceeded
(Hartmann et al., 2023). In addition, runaway $CaCO_3$ precipitation, a condition where more alkalinity is removed
than initially added, reduces the OAE $CO_2$ uptake efficiency. More complex precipitates containing Fe, Si, and P
were observed in a study using a limestone-inspired OAE approach revealing that mineral precipitation caused by
seawater alkalinization can also remove inorganic nutrients from solution (Gately et al., 2023).

Maintaining alkalinity following OAE is critically dependent on the carbonate saturation state, its temporal
evolution, and particle surface processes (Hartmann et al., 2023). To minimize the loss of alkalinity and maximize
alkalinity enhancement, Hartmann et al (2023) propose the application of an alkaline solution in $CO_2$ equilibrium
with the atmosphere and/or solutions with tested saturation levels to prevent a further increase in supersaturation,
and the precipitation of carbonate `to avoid loss of alkalinity. A separate reservoir where alkaline solutions have
been prepared is desirable for testing upper limits of alkalinity addition and identifying saturation thresholds to
minimize precipitation.

**4. Experimental design**

**4.a Experimental replication**
Replication is important to determine if results are reproducible although one must consider that when results are so
dependent on precise experimental conditions that replicability is needed for reproducibility, the result may be
unique and potentially less relevant than a phenomenon that can be reproduced by a variety of independent, non-
identical approaches (see Casadevall and Fang, 2010). A number of experimental designs can be used to achieve
adequate statistical replication (Fig. 1). For example, simple replication involves experimental units (each of the
replicates) per treatment where all the conditions are manipulated independently but in the same way for that
treatment and where responses to the treatment are measured [defined by Hurlbert (2009) as the "evaluation unit"]
and each experimental unit can be considered as independent. In temporal replication, multiple measurements are
made through time (temporal trends) on the same experimental unit. Sacrificial replication involves the use of
multiple sampling times per treatment (for example, a time series) and multiple experimental units at the time of
samplin. Each approach has distinct strengths and limitations, and the choice of the approach depends on the
scientific questions and the extent of the risk of error propagation. For example, one might choose sacrificial
replication for certain chemical manipulations that require sampling from vessels with comparable volumes but
choose instead temporal replication for monitoring the evolution of a microbial culture or the physiology of fish over
time under certain alkalinity conditions.

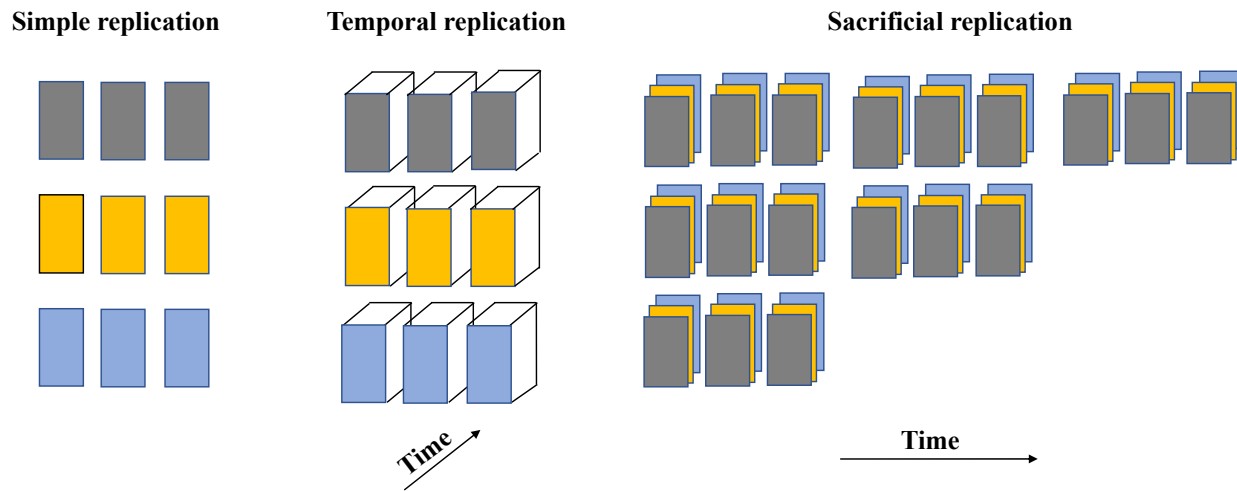

**Figure 1.** Examples of experimental laboratory design with regards to replication. Each treatment, represented by a
colour, contains experimental units contains (replicates). Each experimental unit is treated as an independent
experiment except in the sacrificial replication approach, where each replicate is treated statistically as an
experimental unit.
**4.b Preliminary experiments**
In addition to testing the biological responses to abrupt enhanced alkalinity, marine organisms can be
exposed to enhanced alkalinity conditions after equilibration of seawater $pCO_2$ with that in the air-phase following
alkalinity addition. Ideally, aeration should be maintained to ensure $O_2$ levels required by marine animals and also
maintain stable $pCO_2$ levels in the alkalinity perturbation experiments. Depending on the organism tested (a few
organisms do not tolerate aeration in tanks), aeration might or might not remain for the duration of the experiment
(Table 1). The vessels used in OAE experiments might not be traditional tanks used in aquaria, but rather any type
of container adequate for different type of organisms (e.g., culture flasks for bacteria, conical flasks, carboys for
phytoplankton, open tanks for echinoderms and fish) with air lines to introduce aeration in the media. When running
multifactorial experiments (e.g., temperature and alkalinity), designing an analysis plan and concrete experimental
questions to interrogate can help determine the sample size and minimum number of treatments.
An analog to OAE is the use of lime soda and other alkali to combat acid rain, which has caused deleterious
changes in freshwater ecosystems for more than half a century in northern Europe and North America. To reverse
some of these' changes a number of governmental and nongovernmental teams have applied lime and other
neutralizing compounds to streams, rivers, lakes, and catchments in the most affected or most ecologically valuable
regions (see Clair and Hindar, 2005). Another example is the effects of seawater buffering mainly by addition of
$Na_2CO_3$ addition) utilized by the commercial shellfish industry (e.g., Ragg et al., 2019), which showed a broad
improvement in larval health compared to undersaturated waters.

Standardizing technical details in protocols, sampling, sample processing and analyses are crucial to control for variation introduced by reagents, sample storage and other factors. The collection and curation of metadata associated with each sample are critical for data interpretation, inter-lab comparison and drawing conclusions to move forward with planning field deployments for research purposes. For studies involving more than one level of biological organization; i.e., grazing experiments, competition experiments, particular attention should be paid to designing adequate controls.

The effects of OAE and its interactions with other parameters might differ depending on the duration of the experiments. Indeed, in ocean acidification studies, compensatory metabolic pathways appear to take longer to become established, depending on factors such as the exposure history (Calosi et al., 2013) and phase of the life cycle (Hettinger et al., 2012). In a study testing ocean acidification and warming, biological effects were not detectable in the short term, but were rather manifested over time (Godbold and Solan, 2013). It was suggested that species responses to seasonal variations in environmental conditions might explain these differences that, depending upon timing, can either exacerbate or buffer the long-term directional effects of climatic forcing (Godbold and Solan, 2013).

**4.c Recommended minimum set of variables to report**
To improve comparability between future work, we recommend a minimum set of variables with the understanding that more variables might be added as new results emerge (Table 2). We recommend to measure and report at least the following variables (shown in bold in Table 2).

- At least alkalinity and one more parameter of the carbonate system must be measured to calculate key carbonate chemistry parameters including bicarbonate and carbonate ions, $CO_2$, pH and the saturation state of $CaCO_3$ polymorphs. This information is critical to determine chemical alterations in the dissolved inorganic carbon system as a result of alkalinization.
- Resource availability (e.g., prey, dissolved inorganic nutrients, light) are needed to monitor the growth conditions.
- Particulate organic carbon (POC), nitrogen (PON), phosphorous (POP) are required to learn about trends in biomass production and stoichiometry.
- Basic physiological properties (respiration, photosynthesis) should be measured to inform biogeochemical models and learn about biologically-mediated fluxes of elements.
- Some functional group-specific properties, particularly those involving mineral precipitation (calcification, silicification) and those with environmental effects (e.g., toxin production) and with climate-relevant impacts (nitrogen fixation/denitrification) in context specific cases.
- Size of offspring and fecundity rates can be used as indicators of transgenerational plasticity and adaptation to alkalinization.

Other variables are important in the exploration of specific questions such as how does seawater alkalinization affect biodiversity?; how does metal bioavailability change under increased pH?; what is the role of organic alkalinity in coastal systems? The variables and protocols listed in this chapter is not exhaustive and only provides a proxy sample largely based on the literature on climate impacts on marine systems and ocean acidification.


**Table 2**. **Examples of responses to ocean alkalinity enhancement to be measured in experimental manipulation studies.** Knowledge need (M=medium, H=high; measurement mode (MM=manual mode; S=sensor; SD=sensor in development). A minimum variable set is highlighted in bold. Selected references are provided as examples of protocols.

| Type of response | Variable | Knowledge need | Measurement mode | Protocol reference |
|---|---|---|---|---|
| Chemical and environmental | **Carbonate chemistry parameters $\{[HCO_3^-], [CO_3^{2-}], [CO_2], pCO_2, \Omega\}$** | H | MM, S, SD | Dickson (2010); Bockmon and Dickson (2015) |
| | Dissolved organic matter | M | MM | Marañón et al. (2004); Sharp et al. (1995) |
| | **Dissolved inorganic nutrients** | H | MM | Worsfold et al. (2013) |
| | Resource availability (prey, light) | H | MM, S | Lawrence et al. (2017) |
| | **Particulate organic matter (C, N, P)** | H | MM | Verardo et al. (1990); Hilton et al. (1996); Pujo-Pay and Raimbault (1994); Fu et al. (2008) |
| | Trace metals (in solution and in aggregates) | M | MM | Guo et al. (2022); Hutchins et al. (2023) |
| | Biologically and biogeochemically relevant elements (e.g., Si, Mg:Ca) | M | MM | Brzezinski (1985); de Nooijer et al. (2017) |
| Physiological | **Basic physiology (respiration, photosynthetic, growth rates; morphometric measurements)** | H | MM, S | Iglesias-Rodriguez et al. (2008); Kelly et al. (2013); Farrell et al. (2009) |
| | **Some functional group-specific physiology (e.g., calcification, silicification, nitrification/denitri** | H | MM, S | Cohen et al. (2017); DeCarlo et al. (2019) |

| | | | | |
|---|---|---|---|---|
| | **fication, toxin production**) | | | |
| | Physiological stress [e.g., heat shock proteins, oxidative stress-related proteins, photosynthetic stress (shifts in quantum yield), morphological alterations (e.g., cyst formation) | M | MM | O'Donnell et al. (2009); Moya et al. (2015); Trimborn et al. (2017) |
| | Incidence of pathogens and disease | H | MM | Asplund et al. (2014) |
| Reproduction | Spawning success | M | MM | Liu et al. (2011) |
| | **Size of offspring** | M | MM | Cao et al. (2018); Johnson (2022); Albright et al. (2010) |
| | Sperm motility | M | MM | Esposito et al. (2020); Havenhand et al. (2008) |
| | Epigenetic analysis | M | MM | Li et al. (2018); Lee et al. (2022) |
| | **Fecundity** | M | MM | Maranhão and Marques (2003); Thor and Dupont (2015) |
| | Hatching success | M | MM | Saigusa (1992) |
| Species interactions | Competition for resources | M | MM | Connell et al. (2013); Guo et al. (2022) |
| | Predation and species interactions | M | MM | Greatorex and Knights (2023); Bacus and Kelley (2023); Mitchell et al. (2023) |

| | Synergistic/antagonistic effects of other environmental parameters | M | MM, S | Gerhard et al. (2023); Khalil et al. (2023) |
|---|---|---|---|---|

**4.d Type of experiments**

Laboratory experiments can be designed to both address short term responses and to explore the longer-term adaptation to chronic exposure to enhanced alkalinity conditions. Filtered natural seawater should be used, when possible, in incubations unless artificial seawater is required (for example, when studying the effect of metal concentrations). Short term manipulations involve the use of batch, semi-continuous and continuous incubation experiments. In *batch incubation experiments*, all resources are provided at the beginning of the incubation, without further addition and sampling takes place during a short time period (hours, days, weeks). Only gases and alkali can be added during the course of the experiment. When biological processes are measured, a phase during the life cycle (e.g., larva/adult; vegetative cells/gametes) or growth (healthy, exponentially-growing/resource-limited, stationary growing organisms /senescent organisms) is typically targetted. Sampling is conducted until the nutrients are consumed and beyond if decaying populations are the focus of the investigation.

Given that resources (light, nutrients) are the limiting factor in batch incubation experiments, the organisms are in the exponential growth phase for a limited time period. To expand sampling and replication during the exponential growth phase, resupply of nutrients using a *semi-continuous culturing* approach can prevent food/nutrients from becoming a limiting factor. When the studied organism is phototrophic, one must ensure subculturing (microbial cultures) or appropriate arrangement or organisms to prevent light limitation. The advantage of semi-continuous culturing is that it allows investigating trends over extended time periods, increase replication and higher yield. Generally, the resource is added manually or pumped from the nutrient supply vessel into the culture vessel during exponential growth or when specific conditions are met (e.g., when a certain biomass concentration is reached).

In *continuous cultures*, the rate of addition of nutrient is controlled to maintain steady state cell growth. This system is known as chemostat, where typically, a volume of culture medium is added and the same volume is removed from the growing culture. A challenge with this type of 'bioreactors' is that, over long time periods, they can be more susceptible to microbial contamination and long-term phenotypic and genotypic variance in the cultures (Reusch, 2013).

*Portable incubation experiments* that simulate regional in situ alkalinity deployments are an important step in understanding seawater alkalinization and its impact on marine organisms prior to field testing. This type of incubation experiments, which simulate alkalinity additions under diverse local in situ parameters (e.g., temperature, irradiance, nutrients), can be accomplished using portable incubators onboard research vessels (i.e., deck incubations) or outdoors, at coastal research facilities (Fig. 2).

When studying photosynthetic organisms high-quality light filters should be attached to the acrylic tank to adjust photosynthetically active radiation (PAR) within the incubator (e.g., Fig. 2). To maintain in situ seawater temperatures, an inflow port can supply seawater to the incubator. Effort should be taken to ensure movement of seawater quickly through the incubator to maintain a uniform temperature.

When collecting natural seawater, one must consider how biological interactions (e.g., grazing) could confound results and filter accordingly. Unlike laboratory experiment, that allow for seawater-air phase $CO_2$ equilibration, portable incubation experiments require instantaneous alkalinity additions; thus, careful consideration should be given to the method of alkalinity addition used. When adding liquid alkalinity, e.g., solutions (e.g., 1 M) of NaOH one must consider that flocculation commonly occurs upon alkalinity addition (Subhas et al., 2022). Adding pulverized minerals directly to the treatment vessels is another option although this method may yield incomplete dissolution or slow dissolution (e.g., Fuhr et al., 2022) with undesirable effects including secondary precipitation, particle aggregation and detrimental biological impacts (NASEM, 2022). Some researchers have opted for mimicking mineral dissolution instead (see Gately et al., 2023). As in the traditional laboratory experiments described above, vessels within the incubator should ideally be aerated during experimentation. In addition to

chemical and biological parameters, PAR and temperature data should be collected throughout the experimental
timeframe through discrete sampling or semi-continuously using sensors and data loggers. The best practices
outlined in Box 1 should be adhered to when planning portable incubation experiments. Effort should be taken to
position the incubator in a way that avoids confounding factors such as light contamination (e.g., from the ship).
**5. Sampling and analysis**
Technical variability amongst experimental methods ranging from sampling and sample processing can
propagate through the various steps before analysis; for example, chemical analysis and molecular work/sequencing
can be error-prone (e.g., Catlett et al., 2020). The use of blanks every time sampling is conducted is essential for
detecting contamination originating from the experiment itself or from the adjacent environment (e.g., exogenous
sources such as surface contamination, flagellates in droplets through aeration, etc.). When possible, several barriers
to contamination are recommended (e.g., filters at various points of aeration). Additionally, for samples (other than
those preserved for analysis of alkalinity, dissolved inorganic carbon analysis or pH) that are kept for further
analyses, contaminants that grow during shipping or while samples are being stored can sometimes be reduced by
freezing at -80 °C, when possible, or by using the appropriate preservatives when storing at ambient temperature is
required (e.g., ethanol, paraformaldehyde, glutaraldehyde). Attention should be paid to the material of vessels where
samples and solutions are stored; for example, avoid borosilicate bottles to store nutrients or alkalinity solutions as
silicate can be leached into solution.
Establishing time series prior to the experiment to determine time frames regarding the appropriate length
of the experiment and frequency of sampling is recommended. It is important to establish well defined time
windows for sampling as well as frequency of sampling to capture physical, chemical and biological properties of
the studied system. It is advisable to limit the time of sample storage to minimize observations that might confound
interpretation of results (e.g., reverse weathering during storage) (Subhas et al., 2022).
**5.a Criteria for key parameters**
For the most part, laboratory experiments are aimed at elucidating the physiological performance and
biogeochemical responses of organisms (rather than communities) to physical or chemical alterations in the
environment although responses in ecological fitness could be drawn from laboratory experiments (Table 2).
Importantly, environmental change can affect species differently and interactions between species that are sensitive
to environmental change can function as ecological leverage points through which modest changes in abiotic
conditions are amplified into large changes in marine ecosystems (see Kroeker and Sanford, 2022). These
interactions can be measured as competition, predation, and symbiotic relationships (mutualism, commensalism and
parasitism) that can vary along environmental gradients that cause stress (Stachowick, 2001; Bruno et al., 2003; Ma
et al., 2023).
Criteria for selection of species should include whether the organism is amenable to laboratory
experimentation, the amount of background knowledge on the organism's physiology and biogeochemistry,
ecological importance of the organism, and local and global impacts. Considerations when selecting organisms
should also include geographic origin (e.g., temperate/tropical/polar) and ecosystem type (e.g., benthic vs pelagic).
Special attention should be paid to those species that (1) significantly impact or respond biogeochemically to
chemical changes caused by alkalinity addition (e.g., possibly calcifiers, photosynthetic organisms); (2) keystone
organisms (e.g., corals, salmon, sea stars, toxin-producing phytoplankton); and (3) organisms/functional groups of
known vulnerability to climate change (corals, urchins).
Calcium carbonate producing organisms are particularly interesting because of their known sensitivity to
changes in carbonate chemistry and because any alteration in their abundance or calcification rates could have
implications in the CDR potential of alkalinization. Mineralogical composition of carbonate containing organisms
might possibly be affected by alkalinization. For example, recent meta-analysis of studies exploring the effects of
the carbonate chemistry shifts caused by ocean acidification revealed effects on shell state, development and growth
rate (Figuerola et al., 2021). Biomineralization studies should explore species-specific responses driven by
mineralogical composition (calcite, aragonitic, high/low Mg calcite) of their tests, shells and skeletons.
Environmental and biological control on calcification particularly any changes in the Mg content in calcite driven by
the use of brucite and other minerals potentially adding Mg to calcite must be reported as calcite with a high Mg
content is less stable in aqueous solutions (Ries et al., 2016). Empirical studies have shown that the Mg/Ca ratio of
Mg-calcite producing organisms generally varies proportionally with seawater Mg/Ca (e.g., Ries, 2004; Ries, 2006)
and therefore particular attention should be paid to the Mg content (and solubility) of biomineralized calcite. The
addition to proposed Ca and Mg containing minerals - Ca(OH)$_2$ (slaked lime), Mg(OH)$_2$ (brucite), CaCO$_3$
(limestone) or (Mg,Ca)CO$_3$ (dolomite) - will alter the Mg/Ca ratio of the seawater. An extensive body of literature
reports biogenic and abiotic precipitation of low-Mg calcite when seawater Mg/Ca falls within the calcite stability
field (seawater molar Mg/Ca < 2) and the biogenic and abiogenic precipitation of aragonite and high-Mg calcite
when seawater Mg/Ca falls within the aragonite stability field (seawater molar Mg/Ca > 2) (Ries, 2010). Thus,
modification of local seawater Mg/Ca ratios by OAE has the potential to favor aragonite and high-Mg calcite
organisms if seawater Mg/Ca is increased, and low-Mg calcite organisms if seawater Mg/Ca is decreased. This is an
important area of future OAE research.
Central to OAE laboratory experimentation is our ability to measure any possible stress induced by
alkalinization and learn about underlying mechanisms behind acclimation to the chemical alterations of seawater
caused by OAE. This can be achieved by measuring basic functions (growth rates, size, reproductive success),
sensitivities to alkalinization might be organism-specific and possibly trophic level-specific (e.g., Voigt et al. 2003,
Gilman et al. 2010) although most laboratory experiments do not address the complexity of trophic interactions.
Similarly, measuring adaptation and diversity in acclimation between and within related organisms is a challenge
and the ocean acidification literature revealed how important it is to pay attention to diversity of responses (see
Kroeker et al., 2010).
Stress is often measured as a reduction in organismal performance or fitness caused by environmental
change (Schulte, 2014). In addition to these general physiological or behavioral responses, markers of stress such as
oxidative stress are often used. For example, it is well established that the production of reactive oxygen species
(ROS) can increase due to environmental stress including ocean acidification (Lesser, 2006; Lushchak, 2011). Many
biomarkers are commonly used for studying oxidative stress in marine organisms (Cailleaud et al., 2007; Vehmaa et
al., 2013) and an increase in ROS and superoxide dismutase and catalase activities have been reported in marine
animals under stress (von Weissenberg et al., 2022). Heat shock proteins (HSPs) are also used as molecular markers
of stress because of their abundance, high sensitivity to stress and being ubiquitously expressed (Gross, 2004).
Among all HSPs, HSP70s are the most studied as a strong up-regulation of HSP70 production has been
demonstrated broadly with the exception of *Hydra oligactis* (Bosch et al., 1998), and some Antarctic animals (La
Terza et al. 2001; Place and Hofmann, 2005).
**5.b Measurements of nutrient uptake rates**
The uptake rate of carbon and other nutrients that results in the observed standing stocks of particulate
matter involve many physiological processes that are sensitive to changes in inorganic carbon chemistry and pH
(Matsumoto et al., 2020). Chemical changes following the addition of alkalinity might alter physiological processes
that represent sources (calcification, respiration) and sinks (photosynthesis) of CO$_2$. One should also pay attention to
the reciprocal interactions between these physiological processes and the chemically altered environment as even
minor changes in biological processes, or in the balance between them, can have implications for the CDR potential
and biodiversity.
One of the most unknown effects of OAE is the fate of biological fixation rates of different elements (e.g.,
carbon and N$_2$ fixation rates). Such rates are measured in batch cultures and bioassay (mixed natural community)
incubation experiments (LaRoche et al., 2010). While the objective of culture experiments is to understand the effect
of environmental parameters on the elemental uptake by particular species in a lab, bioassay experiments have to
deal with a rather complex species interaction in the field or after subsampling of mesocosms in a lab (Hutchins et
al., 2007; Paul et al., 2016). Labelled/enriched (~99%) stable isotope tracers is the most used method for rate
estimation these days. The rate calculation is based on isotopic mass balance equation (Montoya et al., 1996):
$$\text{C or N}_2 \text{ fixation rate} = \frac{[POM]}{t}\left(\frac{A_f - A_0}{A_e - A_0}\right) \qquad (1)$$

where, [POM] is the concentration of element of interest (C or N) at the end of the incubation. Likewise, *Af*
= atom% in *POM* at the end of incubation, *A0*= atom% in *POM* at the start of the incubation, *t* is time of incubation,
and *Ae* = isotopic enrichment in the dissolved form after the tracer addition at the start of the incubation

This equation/method is sensitive to analytical protocols in routine incubations (White et al., 2020), and
might be even more sensitive in OAE incubations due to the issue of gas equilibration in tightly capped bottles. While
the C substrate-based incubations are supposedly straightforward in incubations, $N_2$ gas incubations face the challenge
of under-equilibration leading to underestimation of rates. But OAE incubations can produce larger errors in the C
fixation estimates as well. This is because $NaHCO_3$ is generally used as a C substrate. To estimate $^{13}C$ isotopic
enrichment after tracer addition (term in equation 1), a DIC value is normally assumed (as it does not change much at
a given region). But OAE is expected to increase (or fluctuate) DIC during the experimental period, and thus a
measured DIC value should be used in the enrichment factor calculation. Likewise, the $^{14}C$-method, which is widely
used for marine primary production and calcification rate measurements due to its sensitivity (Nielsen, 1952), also
requires treatment-specific determination of DIC concentrations. Likewise, slow dissolution of $N_2$ gas poses a
challenge to accurately estimating isotopic enrichment factor (*Ae*), and it is advisable to measure this term.
Although the analytical precision of C and N isotopes is of order of sub permil levels, many times the low reported
rates (<0.1 nmol N $L^{-1}$ $d^{-1}$) are questionable (Gradoville et al., 2017). Therefore, the detection limit of rate
measurements and its proper reporting is a major concern. To overcome this, following the propagation of analytical
and statistical errors in each term of mass balance equation (1), Gradoville et al. (2017) have proposed to report
minimal quantifiable rates (MQR) and the limit of detection (LOD) in triplicate samples. We ought to follow these
protocols in the rates measured in OAE. In addition, we must make sure to sample/filter sufficient water to achieve
35 µg N and 150 µg C in the sample for reliable mass spectrometric measurements.



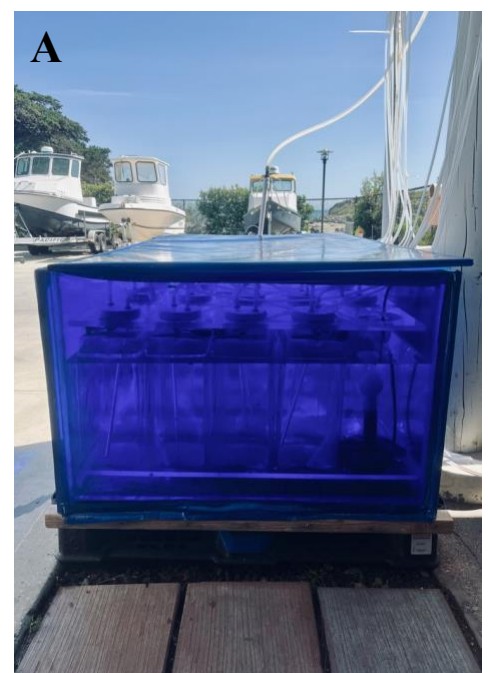
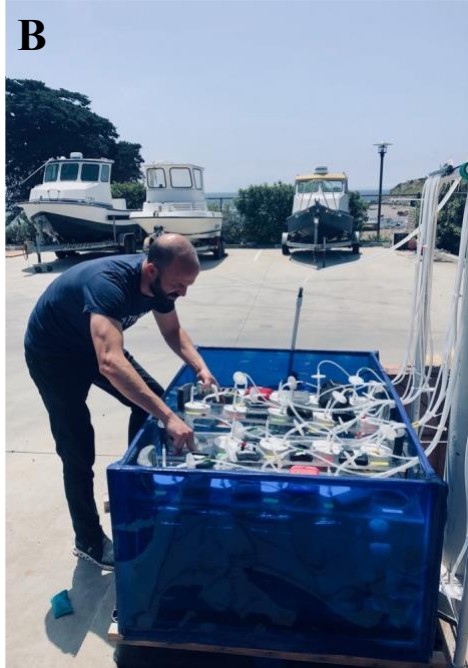
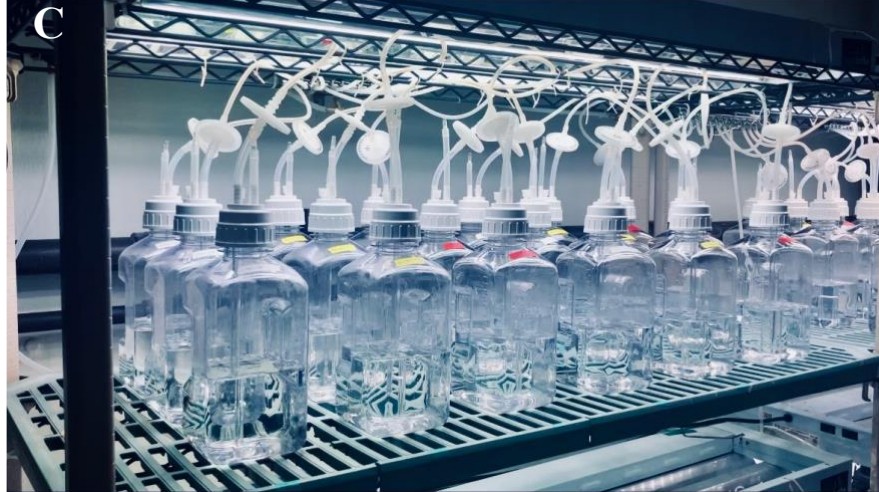

**Figure 2.** A, B: Portable incubator with blue filters to adjust photosynthetically active radiation (PAR). A scalar PAR sensor (LI-COR) can be observed within the incubator (A, right side). C: laboratory experiment using aeration and sacrificial replication. Images were taken by James Gately (A, C) and Sylvia Kim (B).

### 6. Conclusions and recommendations

The field of OAE faces a great diversity of challenges given the continuously evolving experimental approaches and emerging data availability that will undoubtedly provide new information and ideas to optimize best practice in laboratory experimentation. This chapter highlights the need for attention to the design, sampling, performance, and analysis of laboratory procedures used in OAE laboratory experiments. The criteria we present to achieve best practice in laboratory experimentation and design focus on reproducibility, factors affecting CDR potential and organism health (e.g., alkalinity conditions leading to flocculation, aggregation), establishing suitable experimental controls, and identifying the appropriate level of biological organization (physiological, molecular) to study biotic responses to OAE. Key response variables informing on alterations in seawater chemistry following alkalinization, growth of organisms /biomass buildup/reproductive success, and biogeochemically relevant properties (e.g., photosynthesis, respiration, calcification) under elevated alkalinity conditions should be measured and reported. The main recommendations include:

• Ensure reproducibility through appropriate experimental design and replication.
• Determine alkalinity thresholds for the formation of precipitates for each experimental approach and
condition.
• In addition to the proposed alkalinity target values of 3000-4000 µmol kg-1 (Renforth and Henderson,
2017), use concentrations exceeding these recommended values to mimic responses at the site of
deployment/non equilibrium and use intermediate alkalinity levels to identify potential nonlinear responses.
• Establish appropriate experimental design to address questions at specific levels of organization (chemical,
physiological, molecular) and assuming different scenarios (e.g., mimicking impacts at the site of
deployment in a non-equilibrated system versus steady state scenarios in an equilibrated system).
Given the emerging nature of ocean alkalinity enhancement as a research field, this chapter will evolve to
update guidelines as more results become publicly available. Frequent assessments of knowledge acquired from
emerging and future studies and review of best practices are needed to keep the OAE community engaged and
forward thinking.
**Author contribution**
M. Débora Iglesias-Rodriguez wrote the original manuscript with contributions from the authors. Rosalind E.M.
Rickaby, Arvind Sing and James A. Gately contributed with the critical review, commentary and revision of sections
and general comments on the manuscript.

**Competing interests**
Competing interests are declared in a summary for the entire volume at:
\url{https://sp.copernicus.org/articles/sp-bpoae-ci-summary.zip}.

**Acknowledgements**
This is a contribution to the "Guide for Best Practices on Ocean Alkalinity Enhancement Research". We
thank our funders the ClimateWorks Foundation and the Prince Albert II of Monaco Foundation. Thanks are also
due to the Villefranche Oceanographic Laboratory for supporting the lead authors' meeting in January 2023.

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

and its implication for carbon dioxide removal. Marine Chemistry 253, 104251.