# Peer review of "Laboratory experiments in ocean alkalinity enhancement research"

_State of the Planet, 2023_

## Author Response (AR1)

We thank the reviewers for their helpful comments and suggestions. We hope the responses we provide address all the concerns.

**Reviewer 1**

C1. "… the abstract promising 4 very useful things but then I don't find I have been given that information on those 4 aspects once I've read the whole paper.
R1. Each of the four key points highlighted in the abstract is now addressed in the manuscript

C2. "Overall I think the paper could do with a slight rearrangement and focus to provide the reader with more useful information – perhaps something that follows their Table 1 (thus making table 1 a more important visual representation of the paper) and provides the details that's in the legend of that table in the text."
R2. The sections have been rearranged following most of the Reviewer's suggestions. Two sections are now embedded in the text rather than a separate section - Abrupt vs chronic exposures and pros and cons of lab experiments. In section 5 (Sampling and Analysis) we have added 'Isotopic measurements of nutrient uptake rates'. The contents from the former 'Species interactions' section are now embedded in '**Lessons learned from ocean acidification research'.**

C3. Improving clarity re: what the paper sets out to achieve in introduction
R3. The introduction has been re-written and it includes the four key points highlighted in the abstract.

C4. Line 22: change "are intended to be" to "need to be"
R4. This has been amended.

C5. Line 35: "The current focus" of what?
R5. This statement has been removed as it did no longer make sense in the re-written introduction.

C6. Line 38-40: Some of the language could be written more confidently, given this is supposed to be recommendation and best practice. This sentence is a perfect example of this, as it suggests (in a  convoluted way) that the authors don't yet know what the best practice should be. Same for Lines 42 to 45.
R6. The message was intentional as we feel it is necessary to highlight that this guide is a first step of an iterative process. In the new version of the manuscript we provide an additional sentence in the first paragraph of the Introduction clarifying that the document will evolve as the OAE community makes results available. There are however examples through the chapter where we could perhaps make more concrete statements in terms of recommendations.

C7. Line 42: "The rich insights obtained in ocean acidification research…" sentence is out of place here. It should be the start of section 2.
R7. This is now upfront, at the start of section 1.b (**Lessons learned from ocean acidification research).**

C8. Line 49: "Offering guidelines provided in this chapter…" this isn't very good English. The sentence needs re-writing.
R8. The statement will be changed to "The guidelines offered in this chapter…"

**C9. 2. Lessons Learned from ocean acidification research**: This section seems to spend the first paragraph saying how bad acidification research was, in terms of improper reporting, improper replication, or improper conditions (which then are justified in the later sentences of the paragraph). Instead it would be useful to frame OAE in the context of what we have learned from acidification research (and other lab experimental work), pointing towards the OA guide to best practice for a lot of detail on lab and chemistry manipulation information that cannot be covered in this chapter. Highlighting the importance for clear media preparation, consistent manipulation of the chemistry, reporting of this data and experimental design. All of which will then be covered in the following sections. In addition, specific to acidification and OAE, is the second paragraph and third paragraphs about DOM but also the multiple stressors.
R9. We feel we present a balanced perspective using mainly conclusions from synthesis studies with respect to the gaps and inconsistencies in OA research. We have avoided portraying OA research as a 'bad or good' example but

rather we highlight what worked and what did not work based on peer-reviewed work (e.g., the remaining two paragraphs of the section – lines 144-169).

C10. Line 57: "In their study…" who's study?
R10. We have changed to "Cornwall and Hurd (2016) reported that 95% of the experimental work…"

C11. Line 61-64: Pointing out the choice of treatment conditions is useful, but the way it is done here is not helpful. It would be more useful to state that careful consideration is needed when deciding treatment levels. Learning from acidification research there are two aspects resources and question trying to be answered. If there are limited resources then focus on realistic levels, however if the question is about determining thresholds or determining the underlying physiological mechanisms of a response, then pushing beyond those realistic levels might have more value.
R11. This statement has been qualified following the Reviewer's suggestion.

C12. Box 1: This box does not appear to be referred to anywhere in the text other than the last sentence of the portable incubation experiments. If it is meant to be a summary of good standard practise then it needs to be more informative to that point. There is more useful information in Ch4.1 on all of these aspects than is presented in this box.  For instance, Reproducibility – experiments need to be considered fully replicable by reporting methods correctly as well as sampling and reporting on all key parameters (see table 2).
R12. While we could easily remove this box we feel there are elements that are specific to lab experiments. We will make sure we make reference to the box through the text.

C13. Line 84: "… we have learned that similar approaches can…" what is meant by this?
R13. What is meant is that "From the emerging OAE research and the OA literature we (the CDR community, the reader) have learned that similar approaches can lead to conflicting and unresolved outcomes". We are of course happy to change the verbal tense (avoid 'we') if the use of first person is against journal policy.

C14. 3. Seawater media preparation: This section, as well as section 4 and 5 could all do with reworking, especially if the authors decide to follow my suggestion in the overall comment. They need to have focussed take away messages – following the lines of Table 1.
R14. We have followed suggestions and these sections have been re-arranged and are now under section 2 (Seawater media preparation and manipulation of carbonate chemistry).

C15.    4. Sources of alkalinity: As above. Given there is a chapter on this topic, here the authors would benefit from being more specific about how one would manipulate the carbonate chemistry in a lab experiment setting. i.e. the types of alkalinity but also the suitability (pros/cons maybe) for types of experiment (phytoplankton, zooplankton, benthic). Given also the caveat (as explained on line 160) that alkalinity above a certain threshold causes precipitates. Does this need to be avoided? What are the implications?
R15. We have addressed the pros and cons of laboratory experimentation but without being prescriptive about the type of organism or experiment but rather we provide choices of experiments if the focus is an organism versus a community of organisms (portable lab experiment). The formation of precipitates is undesirable in terms of CDR potential and also because of potentially adverse biological and physical impacts. This has been made clear.

C16. 5. Impacts of impurities: As above.
R16. We have followed suggestions re: the structure and clarity when we make recommendations.

**C17. 6. Experimental replication:**
Line 229: replace "several approaches can be applied experimentally to address replication" with "Replication is important in order to… …there are a number of experimental designs that can be set up that allow adequate statistical replication".
R17. This has been amended and a sentence has been added to distinguish between replication and reproducibility..

C18. Line 230: "experimental unit (containing replicates)" is not an experimental unit if it then contains replicates within it! Simple replication involved having more than one experimental unit per treatment where all conditions are manipulated independently but in the same way for that treatment. These experimental units can be considered as independent.

R18. This has been amended both in the text and in the figure.

C19. Line 233: "treated as independent experiment units of a treatment" is incorrect. Temporal sampling of the same unit is not independent as there is more chance the response is similar at points closer in time than further away in time (see section 4.4.3 of Guide to best practice for acidification research, or other stats books). However, as long as we have replicate units being sampled through time then we can use repeated measures statistics to account for this non-independence.

R19. By independent we meant that, aside from the experimental units receiving the same treatment, "the experimental units will not be subject to conditions that are, on average, more similar than are the conditions to which two systems each assigned to a different treatment are subject" (see Kozlov & Hurlbert, 2006). We have deleted "and treated as independent experimental units of a treatment" to avoid confusion.

C20. Second paragraph (line 243-254) is not about experimental replication but instead is about reducing errors and storage issues. This section should be amended and moved to a section about sampling and analysis.

R20. This paragraph is now the first paragraph of section 4 (Sampling and Analysis).

C21. 7. Testing impacts on marine organisms: As per suggestion, parts of this section need to be merged with other sections to form a useful guide to experimental design, and/or sampling and analysis. For instance the first paragraph is about getting the right experimental design (using short term manipulations, or longer term equilibrations), while the second paragraph is about sample collection and preventing errors. The third paragraph here does not provide much useful information in its current form. It should be part of the experimental design considerations for when deciding what type of experiment to choose depending on organism life-history, pre-exposure, and multiple stressors. Probably can be merged with section 8 paragraphs.

R21. Most of this section has been incorporated into a new section on Experimental Design's subsections 4.a (Criteria for key parameters) and section 3c (Type of incubations).

C22. Sections 8, 9, 10 and 12 have useful information buried in them, but would benefit from re-ordering with an experimental design context. Choose species, choose experimental duration – stress responses vs other physiological responses vs longer term responses (such as growth, reproduction). Interaction and community experiments. Portable (ship-based) vs non-portable set ups.

R22. The content of these sections has been rearranged and included in section 4 (Sampling and Analysis section).

C22. Sections 11 and 12 both seem oddly detailed compared to the rest of the sections. Especially section 12, e.g. Line 419 "once the vessels have been placed into the tank, they should be secured…" this is very detailed (unnecessary!) methods compared to most of the other sections on lab set ups.

R22. These sections have been balanced in content and detail with the other chapter sections in the new version of the manuscript. Details have been removed.

C23. Figure 1: The sacrificial replication need to illustrate that there are replicates within each of the units that are sacrificed at each time point – and hence time should also be illustrated here. There should be a distinction between repeated sampling through time within the same experimental unit (temporal replication) and sacrificial replication which has much more initial replicates per treatment in order to make sure there are enough to follow through time (e.g. three time points, three replicates, would need 9 initial units at the start, when 3 are sacrificed at T1, leaving 6, then 3 more sacrificed at T2, leaving final 3 for sampling at T3).

R23. The temporal component of the sacrificial replication has been amended in the figure as well as the number of initial experimental units and the decline in numbers over time.

C24. Table 2. Please add inorganic nutrients to the basic chemistry variables to be measured as well as temperature and salinity. It seems obvious that these should be measured, but given the lessons we learned from acidification research that often these are inadequately reported and therefore not comparable, maybe they need to be explicitly stated here!

R24. These new variables will be added to the table.

**Reviewer 2**

The manuscript by Iglesias-Rodriguez et al provides a useful framework for designing and implementing experiments aimed at investigating impacts of ocean alkalinity enhancement (OAE) on marine organisms. The authors wisely use the framework for ocean acidification (OA) experiments, which emerged over the past couple of decades of intensive research on the subject, as the starting place for their proposed OAE experimental framework.

C1. My main suggestion for the authors as they revise their manuscript is to avoid being too prescriptive and restrictive in their recommendations. As the authors are aware, some of the foundational studies in the field of OA research came from experiments that did not conform to the style of experiments that most other workers were utilizing. For example, Jury et al (2010) conducted OA experiments on corals where they manipulated carbonate chemistry through the addition of various combinations of acid and alkalinity, rather than through $CO_2$-equilibration. Although this approach was not recommended by the OA Best Practices Guide, it allowed for the deconvolution of the impacts of the various aspects of the seawater carbonate system on coral calcification, ultimately leading to the important finding that coral calcification is most strongly influenced by the bicarbonate ion concentration – a finding that would not have been evident if the experiments were conducted via $CO_2$-equilibration as recommended. It also seems overly prescriptive to limit the upper levels of alkalinity addition to such low levels as described in the manuscript (3000-4000 umol/kg-sw). It is always better to have too much information about how a system works, than too little. If workers looking at higher alkalinity levels produce studies that do not fit perfectly into a meta-analysis because other workers did not evaluate such equally high levels of alkalinity, then the worker conducting the review paper or comparative study can simply restrict their comparison to the range of alkalinities that were tested by all studies under evaluation. Like for OA experiments over the past two decades, OAE experiments will not only provide insight into how marine organisms will respond to applied chemical CDR, it will also provide tremendous insight into the source(s) of carbon for photosynthesis and calcification, thereby improving our understanding of the very mechanisms of tissue and shell production in marine organisms. However, to maximize the insight gained from OAE experiment about these more fundamental processes, we should not unnecessarily restrict the scope and design of these experiments so that they are optimized only for applied CDR via OAE, just as we should not restrict OA experiments so that they are only optimized for year 2100 IPCC $CO_2$ emission scenarios.

R1. We agree with the Reviewer and we have modified the last paragraph of section 1b (Lessons learned from ocean acidification research) to address the Reviewer's comment.

C2. Lastly, I would also recommend that the reviewers incorporate information that has been gleaned from quasi-OAE experiments conducted in parallel fields investigating the impact of river liming to combat the effects of acid rain (mainly 1960-1980s) and the effects of seawater buffering (mainly by $Na_2CO_3$ addition) utilized by the commercial shellfish industry.

R2. We have added a paragraph (line 473-479) with the suggested background work for context.

C3. The authors should be commended for their efforts in producing a manuscript that should ultimately prove to be extremely useful in standardizing and advancing the emerging field of experimental OAE research.

R3. We thank the Reviewer for their positive comment.

C4. Line 25: consider adding 'meta-analysis' as reason for standardizing methods

R4. Meta-analysis has been added.

C5. 38: invert latter half of sentence for clarity ('impacts of OAE approaches at various levels of…'

R5. This has been amended.

C6. 45: consider expanding 'interlaboratory comparisons' to include 'interspecies comparisons', both of which would benefit from the standardization of methods advocated for herein.

R6. Interspecies comparison has been included in the sentence.

R7. 53: In addition to the section on 'Lessons learned from OA research', the authors may want to consider including a section on the OAE research that has been conducted to date. There have been numerous studies conducted on the impact of OAE on aqueous organisms from past research by the shellfish industry investigating the utility of so-called 'sweetening' the water through addition of mainly soda ash ($Na_2CO_3$), a practice utilized in shellfish hatcheries for decades, and also in the academic and industrial fields of 'river liming', which dissolved primarily $CaCO_3$ and dolomite in higher latitude watersheds to offset the effects of acid rain (due to NOx and SOx emissions) in the 1960s and 1970, but is still practiced today in Canada and some Scandinavian countries, among

other places. Although the goals of increasing the alkalinity of hatchery waters (for shellfish) and river waters (for offsetting acid rain) was not to sequester CO2, many of these studies did evaluate the impacts of these activities on the organisms – which is the subject of the present chapter.

C7. We have added a paragraph in section 4b (preliminary experiments).

C8. 69: Authors mention that DOM 'contributes' to alkalinity of seawater. But DOM can also reduce (rather than 'contribute' to) alkalinity on a net basis by releasing organic acids. It may be worth mentioning that alkalinity can be both increased or decreased by DOM, depending on what type of DOM is released and what happens to that DOM after release.

R8. This has been amended (see third paragraph of section 1b).

C9. 70: change 'to' to 'in', or 'taken' to 'given'

R9. This has been amended.

C10. 78-79: 'given the capacity to drive shifts..'; need to include reference the subject in this part of the sentence. i.e., given the capacity of what to drive shifts?

R10. This has been amended. "…given the capacity of these parameters to drive shifts…".

C11. 79: consider including 'physiological outputs, survival, and reproduction' along with shifts in species abundance, etc., since those former parameters are the ones that will more often be measured in OAE and OA studies.

R11. These parameters have been included.

C12. 92: consider adding 'and precipitation' after 'dissolution', since both are relevant

R12. This will be amended.

C13. 92: sentence contains a non sequitur

R13. This has been amended.

C14. 101: sodium is a metal

R14. "Metals" has been replaced by 'trace metals'.

C15. 106 : rates of growth and calcification can also be measured by changes in dry mass or buoyant mass in many species, especially macroinvertebrates and macroalgae (see Davies, and others).

R15. These methods have been included in this section and the appropriate references added.

C16. 140: For clarity, consider adding 'and therefore the saturation state of the seawater with respect to CaCO3' to end of sentence.

R16. This has been amended as suggested.

C17. 145: there is also concern about metals leaching from other sources of alkalinity beyond olivine, such as from basalt, brucite, and even carbonates associated with zones of hydrothermal alteration where metals can be enriched.

R17. It has been clarified that metals can leach from other minerals.

C18. 147: NaOH is probably not a scalable source of alkalinity for OAE because its production (the chlor-alkali electrochemical process) requires energy (which releases CO2 if derived from fossil fuel) and, more importantly, would produce massive amounts of HCl as a byproduct — which, from an Earth-system mass balance perspective, would eventually cause ocean acidification and re-release of the sequestered CO2. I would avoid advocating for any particular form of alkalinity in this chapter. The jury is still out on that one -- indeed, this is one of the goals of the forthcoming OAE experiments.

R18. We have modified statements that endorse a particular OAE approach.

C19. 147: just because a waste product (HCl) can be used as a 'cleaning product' does not make it 'clean' in the environmental impact sense. I would not include this line of reasoning here.

R19. This statement has been removed.

C20. 149: NaOH is indeed mined, it is just mined from the sea rather than from land.
E20. This statement will be revised.

C21. 153: specify: a drop in 'aqueous' CO2
R21. This has been specified.

C22. 157: since one of the key goals of OAE research will be to determine the maximum alkalinity that different species can tolerate without deleterious effects, I would strongly recommend against prescribing such a low upper-limit to the OAE experiments as 3000-4000 umol/kg-sw. It is likely that seawater that receives alkalinity or OAE could increase well above 10,000 umol/kg-sw at the instant of addition, and still not cause reprecipitation if diluted/mixed by waves and wind. I would suggest encouraging workers to explore a wide range of alkalinities to maximize inference from their experiments and avoid limiting the applicability of their research in the event that OAE practitioners ends up deploying alkalinity at concentrations greater than 4,000 umol/kg-sw (which they likely will, at least immediately after release). An analogous misstep was also made in the field of OA research, which initially recommended only investigating pCO2 levels within the 400 to 1000 uatm range, as that is what is predicted for the open ocean over the next century, when in reality many of the estuaries, marginal marine environments, and shelf systems that host most of the marine calcifiers in the ocean already experience pCO2 conditions in the 1500-2000 uatm range today due to seasonal eutrophication, upwelling, etc. I would conservatively suggest that OAE experiments investigate alkalinities up to at least 10,000 uatm/kg-sw, which is arguably the maximum that can be maintained without inducing reprecipitation, and – importantly – workers should utilize intermediate treatments (e.g., 2000, 4000, 7000 uatm/kg-sw, rather than just low/high treatments, in order to identify potential nonlinear and even parabolic responses – an important and unexpected outcome of early OA research (e.g., Ries, J.B., Cohen, A.L., McCorkle, D.C. 2009. Marine calcifiers exhibit mixed responses to CO2-induced ocean acidification. Geology, 37 (12):1131-1134.).
R22. We agree with the Reviewer in that it is perhaps too early to make a recommendation for upper limits of alkalinity addition. Although there is evidence suggesting the formation of secondary precipitates with alkalinity values <5000 umol/kg seawater, engineering research on mechanisms to deploy alkalinity to minimize precipitation will be crucial to define upper limits. Our recommendations are now sufficiently broad to promote learning about the system.

C23. 166: consider specifying at the onset of this section that the formulation of f/2 media, or other nutrient media, would only be applicable to experiments on algae.
R23. This has been specified.

C24. 213: considering using 'carbonate precipitation' instead of 'carbonate formation' to differentiate the process of carbonate ion formation from carbonate mineral formation.
R24. We have changed to 'carbonate precipitation' from 'carbonate formation'.

C25. 217-219: sentence should be rephrased for clarity
R25. This sentence has been removed and a statement on the importance of sample storage is now included in a new section on 'experimental design and sampling'.

C26. 224: sentence would benefit from clarification of the term 'tested saturation levels'.
R26. This statement has been clarified.

C27. 243: 'sample' instead of 'sampling'
R27. This has been amended.

C29. 302: sentence should also include a statement that prior empirical studies have shown that the Mg/Ca ratio of Mg-calcite producing organisms generally varies proportionally with seawater Mg/Ca (e.g., Ries, J.B. 2004. The effect of ambient Mg/Ca on Mg fractionation in calcareous marine invertebrates: A record of Phanerozoic Mg/Ca in seawater. Geology 32(11):981-984; Ries, J.B. 2006. Mg fractionation in crustose coralline algae: Geochemical, biological, and sedimentological implications of secular variation in the Mg/Ca ratio of seawater. Geochimica et Cosmochimica Acta 70:891-900) – which is key to the logical argument that adding Mg via brucite

addition (or Ca via CaCO3 or Ca(OH)2 addition) would locally modify seawater Mg/Ca and, therefore, the Mg content (and solubility) of biomineralized calcite.

R29. A statement including this work has been added.

C30. 303: The Bischoff et al (1987) reference applies mainly to synthetic abiogenic calcite. For a recent reference to support the statement that biogenic calcite dissolves more quickly (i.e., is less stable) when it contains higher Mg/Ca ratios, see: Ries, J.B., Ghazaleh, M.N., Connolly, B., Westfield, I., Castillo, K.D., 2016, Impacts of ocean acidification and warming on the dissolution kinetics of whole-shell biogenic carbonates. Geochimica et Cosmochimica Acta 192: 318-337. doi: 10.1016/j.gca.2016.07.001.

R30. We have included the more appropriate reference to support our statement.

C31. 303: I would recommend expanding this section to include discussion of the impacts of the major metals ($Mg^{2+}$, $Ca^{2+}$) associated with the alkalinity source on marine calcifiers. For instance, alkalinization with the common alkalinizing minerals Ca(OH)2 (slaked lime), Mg(OH)2 (brucite), CaCO3 (limestone) or (Mg,Ca)CO3 (dolomite) will modify the Mg/Ca of the affected seawater, which has been shown by an extensive body of literature to favor both the biogenic and abiogenic precipitation of low-Mg calcite when seawater Mg/Ca falls within the calcite stability field (seawater mMg/Ca < 2) and the biogenic and abiogenic precipitation of aragonite and high-Mg calcite when seawater Mg/Ca falls within the aragonite stability field (seawater mMg/Ca > 2) (c.f., Ries, J.B., 2010, Geological and experimental evidence for secular variation in seawater Mg/Ca (calcite-aragonite seas) and its effects on marine biological calcification. Biogeosciences 7: 2795-2849.) Thus, modification of local seawater Mg/Ca ratios by OAE has the potential to favor aragonite and high-Mg calcite organisms if seawater Mg/Ca is increased, and low-Mg calcite organisms if seawater Mg/Ca is decreased. This goes beyond the issue of higher-Mg calcite shells being more easily dissolved in seawater than lower-Mg calcite shells that is addressed in this paragraph, and would probably be worth highlighting as an important area of future OAE research.

R31. We have included a brief discussion on the potential repercussions of altering the Mg/Ca ratio on the selection of calcifiers bearing different polymorphs of $CaCO_3$.

C32. 389: 'propagation'
R32. The spelling has been corrected.

C33. Table 2: the references cited for seawater Mg:Ca manipulation studies (Brzezinski 1985 etc) do not include protocols for experimental manipulation of seawater Mg/Ca. A more relevant reference is needed.

R33. A reference for experimental manipulation of seawater Mg/Ca has been included. Brzezinski, 1985 makes reference to Si analysis.

**Reviewer 3.**

General
The article is a necessary addition to the guidelines for OAE experimental research. It is well written and generally covers the diversity of experimental targets (i.e. organisms) and treatments required for a thorough understanding of OAE biological responses. However, at times the article either needs to recognize this need to cover all organisms and not focus on specifics, or identify that the recommendations presented are more relevant to certain groups (e.g., Table 1). At times the article could also be stronger in its recommendations; reproducibility, representative controls, clear methodology and recognizing inherent limitations are key themes for the article and at times these could be more strongly emphasized to ensure the reader follows them in their future endeavors.

Specific comments
C1. Ln 57: Please make clear that 'their study' refers to Cornwall and Hurd (2016), and also what does 'interdependent' refer to (replication or treatments)?
R1. We have amended this line in the revised manuscript. "Interdependent" refers to the method of replication. An example of interdependent replication is given in Cornwall and Heard (2016): "An example of replicates within treatments that are interdependent are treatment replicates that all share a common header tank that is not shared with replicates of other treatments…". They further note that interdependence can be avoided in experimental design by randomly interspersing replicates of a treatment.

C2. Lns 73-80: Resource availability (nutrients, light, prey) is another confounding factor that it would be good to mention here as this will determine the strength of response. It is also an important difference between laboratory and field experiments that needs consideration when scaling from one to the other. This is followed up later in the article but here it would be good to introduce these issues.
R2. We have incorporated the Reviewer's suggestion in our revised manuscript.

C3. Ln 109: Removing 'as controls' from this line would avoid confusion with untreated controls.
R3. We has been amended this in the revised manuscript.

C4. Ln 116: Suggest adding the caveat that where sterilization is not required for the experimental set up.
R4. We have incorporated the Reviewer's suggestion into the revised manuscript.

C5. Ln 153: Do the authors consider that determining responses to the 'initial spikes in pH and drops in CO2' is an important step, or should the focus be on the response under 'steady state via bubbling'?
R5. Determining responses to initial alkalinity spikes is an important step in OAE research in addition to determining steady-state responses. We have clarified this in the revised manuscript.

C6. Ln 165: Medium preparation is necessary in terms of experimentation with microalgae, what considerations are needed for other organisms (e.g., benthic organisms, zooplankton, fish)?
R6. We have clarified this statement and include considerations for preparing medium for other organisms.

C7. Ln 385: Whilst the authors focus on 13C methodology for rate determination, would it be advantageous to mention here that treatment-specific determination of DIC concentrations is also recommended for 14C methodology (e.g., in studies looking at photo-physiology, where P v E methods are required, or calcification rates)?
R7. We have included this clarification.

C8. Ln 391: This seems like a weak recommendation ('we ought to follow these protocols ..') and stronger wording is suggested.
R8. We have removed this sentence and, when appropriate, we use more explicit statements in the revised manuscript.

C9. Ln 422: What about light contamination from the ship – this is an important additional consideration.
R9. We have noted that effort should be taken to position the incubator in a way that avoids confounding factors such as light contamination. This has been added at the end of the section 3c (Type of experiments).

C10. Ln 450: Shouldn't resource availability (prey, nutrients) not be a key variable to measure during the experiments?
R10. Species interactions (e.g., predation, competition for resources) are included in Table 2. However we have listed 'Resource availability' as a Chemical and Environmental type of response.